# Toward the Development of Epigenome Editing-Based Therapeutics: Potentials and Challenges

**DOI:** 10.3390/ijms24054778

**Published:** 2023-03-01

**Authors:** Jun Ueda, Taiga Yamazaki, Hiroshi Funakoshi

**Affiliations:** 1Department of Advanced Medical Science, Asahikawa Medical University, Asahikawa 078-8510, Hokkaido, Japan; 2Division of Biomedical Research, Kitasato University Medical Center, Kitasato University, 6-100 Arai, Kitamoto 364-8501, Saitama, Japan

**Keywords:** epigenome editing, epigenetics, chromatin, chromatin plasticity, in vivo, drug delivery

## Abstract

The advancement in epigenetics research over the past several decades has led to the potential application of epigenome-editing technologies for the treatment of various diseases. In particular, epigenome editing is potentially useful in the treatment of genetic and other related diseases, including rare imprinted diseases, as it can regulate the expression of the epigenome of the target region, and thereby the causative gene, with minimal or no modification of the genomic DNA. Various efforts are underway to successfully apply epigenome editing in vivo, such as improving target specificity, enzymatic activity, and drug delivery for the development of reliable therapeutics. In this review, we introduce the latest findings, summarize the current limitations and future challenges in the practical application of epigenome editing for disease therapy, and introduce important factors to consider, such as chromatin plasticity, for a more effective epigenome editing-based therapy.

## 1. Introduction

Genome-editing technologies, such as zinc-finger nuclease (ZFN), transcription activator-like effector (TALE) nuclease (TALEN), clustered regulatory interspaced short palindromic repeats (CRISPR)-associated protein (Cas) (CRISPR-Cas), and other related technologies, are rapidly developing. Combined with DNA-cleaving enzymes (such as Fok I and Cas proteins), which have RNA or protein domains that recognize target DNA sequences, these technologies more readily facilitate the repair or modification of target genes or genomic loci than do conventional gene-targeting methods [1,2,3,4,5]. In addition, these technologies are optimized to induce changes in target DNA sequences, which make them ideal for repairing genomic DNA or introducing mutations. Studies on the application of genome-editing technologies for treating human genetic disorders and infectious diseases, such as human immunodeficiency virus (HIV), are ongoing [6]. Genome-editing therapies have just begun clinical trials, and approved drugs have not yet been commercialized [7,8,9,10]. For genome editing to be applicable to humans, DNA repair must be complete, and the repaired genome must be error-free. However, these goals are currently difficult to achieve through the intrinsic DNA repair mechanism, as the DNA repair pathway is complex [11].

An alternative approach to regulate gene function is to rewrite the epigenetic landscape to control gene expression with no or minimal changes in the underlying DNA sequence. Epigenome editing is considered a potential therapeutic approach for various genetic diseases and certain cancers [12,13]. These genetic diseases mainly involve gain-of-function genes [14,15], as loss-of-function genetic diseases can be alternatively treated via conventional gene therapy [16]. Epigenome editing is also considered for the treatment of nonhereditary diseases (or nongenetic diseases), as the target gene expression can be increased or decreased by modifying the epigenome associated with the target gene [17,18]. Moreover, epigenome editing is a practical approach to artificially manipulate the chromatin structure of arbitrary genomic regions, thereby enhancing our understanding of the basic epigenomic research [19]. This review outlines the fundamentals of epigenetics and epigenome editing and focuses on the current status and potential of epigenome editing, including ethical considerations, in the treatment of various diseases.

## 2. Overview of Epigenetics and Epigenome Editing

All cells in the human body have the same genetic information. However, during cell differentiation, dynamic chromatin remodeling sorts the necessary genes from unnecessary genes. The changes in chromatin structure ensure that the necessary genes are actively transcribed, and the unnecessary genes are not transcribed because of the formation of heterochromatin structures [20,21,22]. These structures and phenotypes are maintained after each cell division. This process is called epigenetics, and intimately involves the chemical modifications of DNA, histones, chromatin proteins, and noncoding RNAs (e.g., small interfering RNAs (siRNAs), microRNAs (miRNAs), PIWI-interacting RNAs (piRNAs), and long noncoding RNAs (lncRNAs)) [23,24,25]. Except during DNA recombination in immune cells, the genomic DNA sequence of most cell types is retained, regardless of differentiation fate. In particular, when histone tails are acetylated, the chromatin structures become relaxed and accessible to RNA polymerases and basic transcription factors that activate the transcription of the gene [26]. On the other hand, when DNA is methylated, or histone tails undergo repressive post-translational modifications, the chromatin is condensed and inaccessible to RNA polymerases and other transcription factors; thus, gene expression is repressed [21].

For instance, lysine and arginine residues of histones H3 and H4 are known to be methylated, exhibiting different results depending on the site of modification (Figure 1). Specifically, methylation of the 4th (H3K4) and 36th (H3K36) lysine residues of H3 is closely related to transcriptional activation and elongation, respectively. In contrast, the methylation of the 9th or 27th lysine residue (H3K9 or H3K27) of the same H3 is involved in transcriptional repression and heterochromatin structure formation [27,28,29]. These differences are caused by the different “reader” proteins that recognize the methylation sites. Therefore, the chromatin state of the modification site is determined by the type of “reader” protein that binds during methylation and post-translational modification. Although the individual chemical modifications involved in epigenetics are reversible, the chromatin structure (the so-called “epigenome”) changes are often stable and robust once the cell has established its identity, largely due to the influence of the “reader” proteins [21,25,30]. Thus, the “reader” proteins ultimately determine the chromatin state. Specifically, the chromodomain (CD) of the heterochromatin protein 1 (HP1) recognizes H3K9 methylation [31], which is observed in the nuclear structure after the phase separation [32,33].

Epigenomic-modifying enzymes, such as histone acetyltransferases, DNA methyltransferases (such as DNA methyltransferase 3 (DNMT3) alpha (DNMT3A)), and ten-eleven translocation methylcytosine dioxygenase 1 (TET1), which is a methylcytosine dioxygenase that demethylates methylated DNA, play pivotal roles in altering the epigenome. However, with a few exceptions, such as PR/SET Domain 9 (Prdm9) [34], most epigenomic-modifying enzymes do not exhibit genome sequence-binding specificity, as they do not have a DNA-binding domain that determines a target sequence. Instead, their target specificity depends on the type of transcription factors with which they form complexes [35,36,37,38]. As transcription factors bind to several target genes (often hundreds to thousands) and the length of the target sequence of an individual transcription factor ranges from a few to a dozen bases [36,39,40], it is difficult to artificially turn on or off the expression of any gene of interest using an epigenome-editing enzyme with a transcription factor DNA-binding domain. Therefore, the DNA-binding domain of a transcription factor is not suitable for targeting epigenomic-modifying enzymes. Hence, for epigenome editing, a programmable RNA or protein domain that recognizes target DNA sequences (in most cases, a target sequence of approximately 20 bp) and the catalytic domain from epigenomic-modifying enzymes (so-called “EpiEffectors”) are fused to create an artificial enzyme [41,42] (Figure 2). By using these artificial enzymes, the EpiEffector can specifically modify the epigenome of the target site. In the next section, we will describe the different types of EpiEffectors which are used in epigenome editing.

## 3. Description of EpiEffector Molecules

EpiEffectors are the enzymatic domain of a group of enzymes involved in epigenetic modifications of DNA and histone proteins that do not themselves bind to specific DNA sequences. Typical examples of EpiEffectors are shown in Table 1. The herpes simplex virus-encoded protein VP16 is involved in the post-translational modification (acetylation) of histone tails and contributes to transcriptional activation. TET1 demethylates methylated DNA, which is a well-known to repress transcription. In contrast, some EpiEffectors, including DNA methyltransferases, are responsible for transcriptional repression. Nevertheless, the chromatin structure is altered, but the DNA sequence is retained in both cases. Although several enzymes are involved in epigenetic regulation (Table 1), only a few EpiEffectors are currently used for epigenome editing in vivo (Table 2). As described in the next section, VP16 and VP64 (four VP16s) are commonly used as EpiEffectors that activate transcription, whereas Krüppel-associated box (KRAB), the transcriptional repressor domain of Kox-1 (also known as ZNF10); a mammalian de novo DNA methyltransferase, DNMT3A; and a bacterial DNA methyltransferase, MQ1, are commonly used for gene repression. In addition, attempts have been made to combine DNMT3A with DNMT3-like (DNMT3L), a stimulator of the catalytic activity of de novo DNA methyltransferases, to extend the duration of epigenome editing in the cell [70,84,85]. Furthermore, epigenome modification approaches, such as the use of synergistic activation mediators (SAM), CRISPRon, and CRISPRoff (single artificial genes containing multiple EpiEffectors), which simultaneously express multiple types of EpiEffectors, are more effective than those that express a single type of EpiEffector [54,70,97]. These EpiEffectors were selected based on their efficacy in vivo, their well-understood properties, and their small gene size favoring in vivo gene transfer (Table 2). For further details on the types of EpiEffectors in epigenomic-modifying enzymes, please refer to earlier reviews [37,41,43,44,45].

## 4. Description of Epigenome-Editing Methods

Early studies using epigenome-editing enzymes with target sequence specificity were performed using zinc-finger and TALE domains that were originally used for genome editing [37,41,43]. Both are artificial enzymes that fuse the EpiEffector molecule with a DNA-binding domain that recognizes the target sequence. Several examples of epigenome editing at the cellular level have previously been reported [41,44], whereas those at the organismal level have been increasing in recent years (Table 2) [12]. In particular, Garriga-Canut et al. used zinc-finger and KRAB to specifically repress the mutant huntingtin gene (*htt*) in the brain tissue from an animal model of Huntington’s disease [14]. Zeitler et al. also used zinc-finger and KRAB to specifically suppress the mutant *htt* gene in cells derived from patients with Huntington’s disease, as well as in an animal disease model [15]. In contrast, Yamazaki et al. fused the gene encoding for the CpG methyltransferase from *Mollicutes spiroplasma* (*M. SssI*, strain MQ1) with TALE and successfully methylated repeat sequences in the pericentromeric region of chromosomes in early mouse embryos [42,87]. This study demonstrates the possibility of epigenome editing in a wide genome region (i.e., genomic DNA size that can be observed under an optical microscope). As transposon activation is a problem in the xenotransplantation [105], inactivation through epigenome editing of numerous repetitive sequences and transposons in the genome may become an important research area in the future. Although these approaches using zinc fingers and TALEs have some advantages (discussed in later sections), they have not yet been widely adopted by the scientific community because of the long period of time required to synthesize target sequence recognition domains [106].

In addition to zinc-finger and TALE systems, CRISPR systems have been developed using dead Cas (dCas) proteins that do not cleave DNA (“dead,” as the Cas protein has lost its endonuclease activity), but possess a programmable DNA-binding activity. The CRISPR-Cas system uses RNA for target site recognition, which makes it easier to recognize target DNA sequences compared with zinc-finger and TALE systems. The advent of these systems has changed the landscape of genome editing [45,106,107]. Among the dCas proteins, dCas9 was first explored for the CRISPR-Cas system, and methods were developed to manipulate the epigenome using dCas9 to achieve various effects and actions. The following is a brief description of the applications of the CRISPR/dCas system in epigenome editing.

First, the direct effector fusion approach uses the effector-fused dCas9 to interfere with transcription through sterically inhibiting RNA polymerase binding and transcription elongation [108,109]. This strategy has been successfully applied in prokaryotes, which reduced the mRNA expression approximately 300-fold when targeting dCas9 using a single guide RNA (sgRNA) and up to 1000-fold when two sgRNAs are combined to inhibit transcription elongation [108,109]. However, in mammalian cells, only an approximately two-fold reduction in transcription levels was achieved [109]. For dCas9 to potently regulate gene expression in mammalian cells, specific effectors, such as transcriptional activation (VP64 and P65) and repression domains (KRAB and Sin3a-interacting domain (SID)) are required. Either of these active or inhibitory domains is then genetically fused with dCas9 to produce a single functional recombinant protein [110]. The dCas fusion proteins that activate or repress transcription are called CRISPR activation (CRISPRa) [111,112,113] or interference (CRISPRi) proteins, respectively [109,113,114]. The second approach, called indirect effector recruitment, incorporates an additional effector protein-recruiting motif into the basic design, of which the SUperNova tag (SunTag) is a representative example [115]. The third approach, called spatiotemporal control of activity, uses split-dCas9 or split-dCas9-effector proteins [116]. In this approach, DNA-binding complexes assemble and function under various conditions, such as chemical or light induction [117,118].

Using these approaches, the therapeutic applications of epigenome-editing have been studied in animal models of genetic diseases (Table 2) [45]. In particular, Matharu et al. successfully treated haploinsufficiency disease in mice through increasing target gene expression to normal levels using *Streptococcus pyogenes* dCas9-VP64 [100]. Thakore et al. also combined *Staphylococcus aureus* dCas9 with the transcriptional repressor KRAB to suppress the expression of the target gene *Pcsk9*, which regulates cholesterol levels, in the liver. In this study, the effect of a single dose of dCas9-KRAB lasted for up to 24 weeks [67]. Furthermore, Horii et al. successfully used dCas9-SunTag and single-chain fragment variable (scFv)-TET1 antibody to generate animal models of Silver–Russell syndrome, which is a disease related to genomic imprinting disorders [93]. More recently, Bohnsack et al. used dCas9-P300 to activate the activity-regulated cytoskeleton-associated protein (*Arc*) expression and observed the attenuation of adult anxiety and excessive alcohol use disorder in rats [55].

When considering the application of platforms used in genome editing to disease treatment, CRISPR-Cas systems are uniquely DNA- or RNA-based therapies because of the use of guide RNA, whereas epigenomic-modifying enzymes based on zinc fingers or TALEs are applicable as protein drugs and can be administered similarly to available commercial drugs. As protein-drug immunogenicity has been studied more widely than gene therapies, the availability of zinc fingers and TALEs as protein drugs could be a major advantage in developing epigenomic-modifying enzymes as therapeutic drugs [119,120]. Moreover, epigenome editing technologies using CRISPR-Cas and TALE are potentially immunogenic in that they contain non-human materials [121]. In contrast, zinc-finger-based technologies have the advantage of being less immunogenic than CRISPR-Cas and TALE because they are composed of polypeptides encoded in the human genome. In any case, one of the challenges for the future will be determining how to reduce the immunogenicity of the components of epigenome editing technologies.

The main feature of epigenome editing, which is the preservation of the nucleotide sequence, has been considered to be both a weakness and strength, as the disease-causing gene is not altered. Instead, epigenome editing suppresses the expression of disease-causing genes or increases the expression of checkpoint genes, such as cell cycle and other suppressed genes. Therefore, except for the different target specificities, epigenomic-modifying enzymes are similar to the available molecularly targeted drugs against epigenomic-modifying enzymes, such as the DNA methyltransferase inhibitor, azacitidine (trade name Vidaza), and the histone deacetylase inhibitor, vorinostat (trade name Zolinza) [122,123]. Molecular drugs that target epigenomic-modifying enzymes deliver therapeutic efficacy by inhibiting the activity of specific enzymes. As they inhibit all functions involving the enzyme, they affect the entire genome (i.e., “epigenome remodeling” of treated cells) and result in substantial side effects in the patient [124,125]. In contrast, epigenome remodeling via an epigenome-editing strategy may result in fewer adverse effects, as it targets only one or a few sequences within the genome and does not act on the epigenomic-modifying enzymes. Furthermore, the modified chromatin structures are maintained after cell division, as they employ the endogenous epigenetic maintenance mechanism within the cells [12]. Consequently, epigenome editing may be suitable for the treatment of dominant genetic diseases, such as those caused by gain-of-function type mutations. Aside from epigenomic-modifying enzymes, miRNAs and siRNAs that target RNA transcripts are being developed as drug candidates for the treatment of genetic diseases, some of which have begun clinical trials for the treatment of dominant genetic disorders, such as Huntington′s disease [126,127]. Another advantage of epigenome editing is its reversible effects compared with the irreversible DNA sequence changes in genomic editing. Accordingly, as with existing drugs, such as molecularly targeted drugs, the dosage, duration of administration, and other factors in epigenome editing may be adjusted according to the patient′s condition.

## 5. Challenges in Epigenome-Editing Technologies

In the treatment of various diseases, including hereditary diseases, epigenome editing is applied to regulate the transcription of target genes without causing substantial side effects [17,18]. To achieve this, five factors must be considered: the off-target effects, undesired genomic mutations caused by the treatment, nuclear structure, cell types, and method of administration. This section summarizes the status of these challenges and possible approaches to overcome them.

### 5.1. Target Specificity in Epigenome Editing

In recent years, the problem of specificity in epigenome editing has been gradually addressed. Although the risks are small, potential problems must be carefully minimized for a successful clinical application. Unlike genome editing that targets coding regions (e.g., exons), epigenome editing targets the transcriptional regulatory regions with similar sequences in several genes, thereby making it relatively difficult to find sequences unique to a specific gene [38,128]. Moreover, most genes are simultaneously and synergistically regulated and controlled by a common set of transcription factors during development and differentiation [36,38,128,129]. In particular, the transcription factors OCT4 and NANOG synergistically form the inner cell mass of the blastocyst [130]. Together with SOX2, these transcription factors regulate several thousands of genes in mouse embryonic stem (ES) cells [36]. This suggests that DNA-binding protein domains and RNAs that recognize target DNA sequences should be designed for longer target sequences when used for epigenome editing. Therefore, instead of the widely used type II CRISPR-dCas9 system [1], the type I-E Cas3 complex, which recognizes a 27 bp target sequence (longer than that in Cas9), may provide advantages in terms of specificity in epigenome editing [131,132,133]. This may be especially useful when applying the dCas3 system to ex vivo studies. However, when EpiEffector targets a transcriptional regulatory region of about 20 nucleotides, it should be noted that the effect of epigenome editing may become local, and the gene may be transcribed normally through alternative splicing and alternative promoter mechanisms [134]. Accordingly, the possibility of such splicing should be considered when applying this technology.

### 5.2. Avoidance of Undesirable Genomic Mutations Caused by Epigenome Editing

Although CRISPR/dCas systems are widely used for epigenome editing because of their easily designed target sequences, the potential for dCas systems to alter genomic sequences remains a concern [135,136]. Laughery et al. reported dCas9 binding and R-loop formation as the main causes of background mutations caused by the dCas9 system. dCas9-induced mutations were particularly prominent when targeting the antisense strand of a gene. Several of the induced mutations resulted from cytosine deamination events induced by dCas9 on the nontarget strand of the R-loop, whereas the other mutations were related to homopolymer instability or translesion DNA synthesis. The results indicate that DNA binding by dCas9 is mutagenic, which is possible because dCas9 induces the formation of R-loops at its target sites [135].

As TALEs and zinc fingers only bind to DNA and have no enzymatic activity to cleave DNA, another cause of background mutation unique to the dCas system could be the residual DNA cleavage activity of dCas proteins. Therefore, if dCas proteins are to be used for human epigenome editing, their DNA-cleaving activity must be eliminated.

### 5.3. Importance of Nuclear Structure in Epigenome Editing

The plasticity of the nuclear structure (genomic organization) of a cell is lost as cell differentiation occurs, and certain diseases are known to have abnormal nuclear structures [137,138,139,140]. During differentiation, the cells change their nuclear structure and translocate transcribed genes (i.e., genes that are used by the cell) to the euchromatic region near the center of the nucleus. Genes that are not transcribed (i.e., genes that are not used by the cell) are translocated near the nuclear periphery, where they become part of the heterochromatin and consequently, are inaccessible to the molecules necessary for transcription, such as RNA polymerases and basic transcription factor machinery. DNA and the histone proteins that fold DNA undergo various chemical modifications, which are eventually recognized by different “reader” proteins, as described previously. Once formed, the chromatin and nuclear structures are maintained and difficult to reverse.

In epigenome editing, the epigenomic-modifying enzymes must be applied in cells with loose nuclear structure and dynamic and plastic chromatin, such as in early embryos and stem cells. This is because nuclear structures and chromatin are likely to be fixed in differentiated cells (Figure 3 and Figure 4A) [137,138,139,141,142], and a single epigenomic-modifying enzyme alone is insufficient to alter nuclear structures and chromatin [47,59]. Furthermore, the simultaneous introduction of several factors is necessary to change the expression state or epigenome of a gene of interest [47,54,70]. In addition, the nucleosome structure may affect the epigenome editing [143]. Specifically, heterochromatin reorganization requires the cooperation of numerous energy-consuming factors, including ATP-dependent chromatin-remodeling factors, to release the heterochromatin state [144,145]. This indicates that epigenome editing in differentiated cells has a poorer therapeutic effect than that in stem cells and thus, is not sustainable. Repeated dosing or increasing the dose of the epigenomic-modifying enzyme may address this problem. Repeated dosing has been applied in existing drugs, such as those targeting epigenomic-modifying enzymes (e.g., azacytidine). Nevertheless, it is important to consider the chromatin structure of the cells to be treated, as the therapeutic effects (including side effects) of the epigenomic-modifying enzyme may depend on the cell type.

Notably, conventional gene targeting was made possible by the development of ES cells [146]. One of the most important characteristics of ES cells is their high chromatin plasticity [137,141]. Accordingly, chromatin plasticity must be considered in human epigenome editing studies.

### 5.4. Selection of Cell Types to Be Subjected to Epigenome Editing

It is important to increase the target sequence specificity of epigenomic-modifying enzymes. It is also equally important to consider the cell types subjected to epigenome editing (Figure 4A). Specifically, to suppress the expression of a target gene through epigenome editing in the cell where that gene is strongly expressed, it is likely that transcription of the entire chromosomal region where the target gene resides will be activated. Although gene expression is suppressed by a target-specific epigenomic-modifying enzyme, the enzyme may not be sufficient, and the effect may not be sustained. Therefore, when the target gene is in the transcription factory, and its expression is active, a large-scale chromatin remodeling is required to suppress its expression, and a single epigenomic-modifying enzyme without chromatin remodeling may not be sufficient. Conversely, a target gene within an inactivated chromosome and heterochromatic region would be inaccessible for expression. Accordingly, the type I CRISPR-dCas3 system possesses epigenomic-modifying and helicase enzymatic activities and can edit long regions of genomes (0.5–100 kbp). Furthermore, compared with conventional systems, the CRISPR-dCas3 system may be more suitable as an epigenomic-modifying enzyme once the problem of genome size is addressed (Figure 4B) [131,133]. Alternatively, the SunTag system, in which multiple EpiEffector molecules are assembled in a scaffold to amplify the epigenomic-modifying enzymatic activity (Figure 4C) [115], is more effective for epigenome editing than a single EpiEffector domain [90,93,94,99,102,147]. Another approach related to the SunTag system is to combine different EpiEffectors to enhance transcriptional activation or repression, which is shown to be successful in vivo (Figure 4D) [54,58]. Finally, targeting multiple loci within the target gene using epigenomic-modifying enzymes may be effective in modulating the epigenome (Figure 4E) [98,99].

Overall, the chromatin plasticity of the target cells for epigenome editing must be sufficiently high to allow alterations to the chromatin. If the chromatin plasticity of the cells to be treated is lost, several epigenomic-modifying enzymes are necessary; however, this may result in off-target effects. Consequently, cell type must be considered when opting for epigenome editing as a therapy.

### 5.5. Method of Administration

Epigenome editing requires more components than genome editing. This inevitably increases the overall genome size of the gene transfer vector. Accordingly, the methods for delivering epigenome editing-based therapeutics to target cells, tissues, and organs, including miniaturization, must be optimized [17,148,149,150,151,152,153]. The application and function of drug delivery systems via viruses, lipids, compressed DNA nanoparticles, or gold nanoparticles must be improved to deliver engineered epigenomic-modifying enzymes to target cells before their terminal differentiation (specific somatic stem cells in which the chromatin structure is not fully immobilized) [12,150,154,155,156,157,158]. As discussed in the previous section, if epigenome editing can be performed on cells with high genomic plasticity, the treatment may be developed with fewer EpiEffectors, which can potentially address the problem of genome size. In gene delivery, it is also important to consider the method of transferring epigenomic-modifying enzymes into specific cells with higher genomic plasticity.

When epigenome editing is applied to humans in situ, the size of the epigenome-editing system is critical for successful delivery. Among the developed systems, the adeno-associated viruses (AAVs) have attracted significant attention (Table 2). Specifically, the AAV type 2 vector has lost the site-specific insertion into human chromosome 19 via mutation. Site-specific insertion is a major characteristic of AAVs, possibly because the gene encoding for the Rab-escort protein-1 (REP1) is removed from the vector plasmid [159]. In this regard, AAV vectors have attracted particular attention as gene carriers for epigenomic-modifying enzymes. However, although the gene may be incorporated into the chromosome of the transduced cell, the probability and degree of such incorporation are considerably reduced. Furthermore, the expression of the gene of interest can be maintained for a long period of time [12,67,160]. In addition, AAVs are ideal delivery systems because of their low immunogenicity, high serotype abundance, and ability to preferentially infect specific tissues. A limitation of using AAVs for gene delivery is that the suitable gene size for epigenome editing must be less than 4.7 kbp (including promoter regions) [161]. However, for several epigenome editing systems, such as the type I and II CRISPR-Cas, consisting of large genome sizes, AAVs are not suitable gene delivery methods. Several approaches have been developed to overcome this limitation. Specifically, instead of using *S. pyogenes* Cas9, smaller dCas9 orthologues, such as SaCas9, SadCas9, CjCas9, and NmeCas9, and Casφ, originating from large phages, have been developed and shown to be successfully incorporated into AAVs [67,69,100,162,163,164,165,166]. In addition, because the size of EpiEffector molecules is generally large, it will be necessary to downsize and optimize EpiEffector molecules in future studies.

Regarding the size limitation, various systems have been studied to allow the delivery of larger genomes. These include the dual/triple vector, concatamerization/trans-splicing, overlapping, hybridizing, protein trans-splicing, single vectors, and mini-gene strategies [167,168,169]. Another approach to overcome the size limitations is to use split inteins. Split inteins are a pair of naturally occurring polypeptides that mediate protein trans-splicing, similar to introns in pre-mRNA splicing when located at the terminus of two proteins [170]. In 2015, Fine et al. [171] discovered the split-intein, SpCas9, which exhibits a moderate genome editing rate in HEK293T cells compared with full-length SpCas9. In 2016, Chew et al. [172] developed an spCas9-AAV toolbox that retains the gene-targeting ability of full-length SpCas9. This set of plasmids includes the AAV-Cas9C-VPR for targeted gene activation. Split inteins are also used to express base editors. Another approach is nanotechnology-based delivery, such as the use of gold nanoparticles or quantum dots, which have been applied to the CRISPR/Cas9 system [173]. Nanocarriers, such as liposomes, polymers, and inorganic nanoparticles, have also been used for gene delivery of CRISPR/Cas gene-editing systems, which suggests that small particles are a viable alternative for large gene transfer [174,175]. However, when using nanoparticles in humans, the risk of epigenetic alterations must be considered [176]. Other possible methods are ex vivo epigenome editing, in which somatic stem cells or other cells are extracted from the patient, epigenome-edited, and then returned to the body of the patient [12,177]. In this case, it is easier to introduce epigenomic-modifying enzymes into the cells, which may be advantageous, depending on the type of disease.

Collectively, the application and function of drug delivery systems via viruses, lipids, compressed DNA nanoparticles, or gold nanoparticles must be improved to successfully deliver engineered epigenomic-modifying enzymes to target cells before their terminal differentiation (specific somatic stem cells in which the chromatin structure is not fully immobilized) [12,150,154,155,156,157,158]. In addition, since epigenetics is reversible, repeated administration of epigenomic-modifying enzymes may be necessary if the target gene needs to be repressed or activated for an extended period. Accordingly, a dosage form that allows repeated administration is also desirable and should be taken into consideration.

## 6. Future Perspectives

Epigenome editing has several important applications in basic research and offers potential novel treatments for various diseases. Although it is still in its infancy, several experimental studies have demonstrated the capability and promise of this technology. In addition, as the size of EpiEffector molecules responsible for the enzymatic activity in epigenome editing is generally large, it will be necessary to downsize and optimize EpiEffector molecules in future studies.

Several challenges in epigenome editing have been discussed: improving target specificity, selecting optimal cell types for epigenome editing, avoiding undesirable genomic mutations, considering nuclear structure, and selecting optimal administration methods. Once these challenges are addressed, and if highly effective epigenomic-modifying enzymes can be delivered to target cells, epigenome editing poses a huge potential for application in human therapies, such as in improving therapeutic efficacies and extending drug responses. Directing epigenomic-modifying enzymes to target sequences is beneficial for the development of therapeutic agents with a lower risk of side effects than existing drugs, such as molecularly targeted drugs. Thus, epigenomic-modifying enzymes could be a promising option for the treatment of various diseases, including genetic diseases. In addition to epigenome editing of single disease-causing genes, future studies on epigenome editing that stabilizes or alters the entire chromosome structure are also important for the treatment of diseases associated with genome instability and chromosomal structural abnormalities [178]. As epigenome editing is relatively safer than genome editing, especially when targeting transposons or repeat sequences that are present in the genome in thousands of copies, further investigations are necessary.

Finally, because epigenome editing does not involve modification of the genome itself, it is currently considered to have a lower impact on germ cells than genome editing. Thus, epigenome editing has the potential to overcome important scientific and ethical issues of concern with genome editing. However, because of the uncertainties associated with new medical technologies, deliberation is essential on how to clear social and ethical issues and develop safe and appropriate strategies and policies [179].

## Figures and Tables

**Figure 1 ijms-24-04778-f001:**
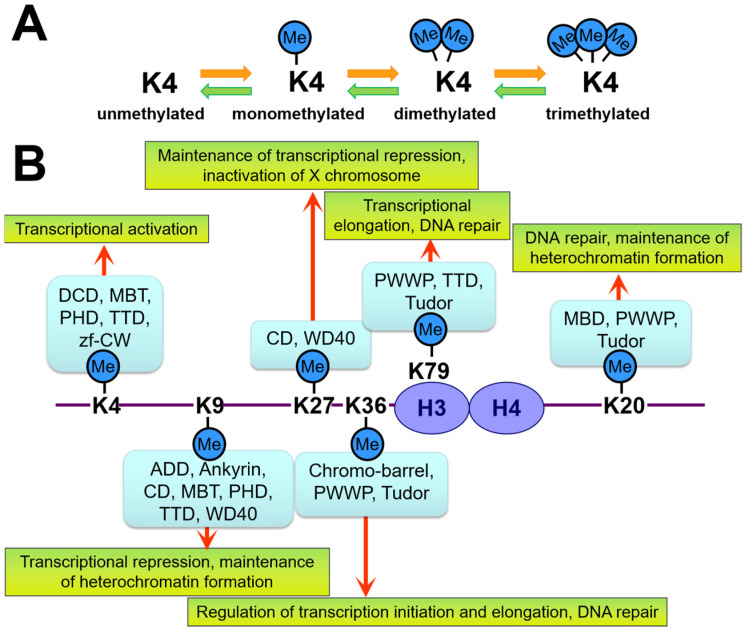
Schematic showing how lysine (K) methylation (Me) is recognized by different domains of “reader” proteins and their outcomes. (**A**) Each lysine residue on the histone can either be mono-, di-, or trimethylated. Each of these post-translational modifications is recognized by different “reader” proteins. (**B**) Methylation of lysine residues of H3 and H4 and the protein domains that recognize them. The binding proteins, and not histone modification, change the chromatin structure. ADD—Alpha-thalassemia intellectual disability syndrome X-linked (ATRX)-DNMT3-DNMT3L; CD—chromodomain; MBD—methyl-lysine-binding domain; MBT—malignant brain tumor; PHD—plant homeodomain; PWWP—conserved Pro-Trp-Trp-Pro motif; TTD—tandem Tudor domain; zf-CW—zinc-finger CW.

**Figure 2 ijms-24-04778-f002:**
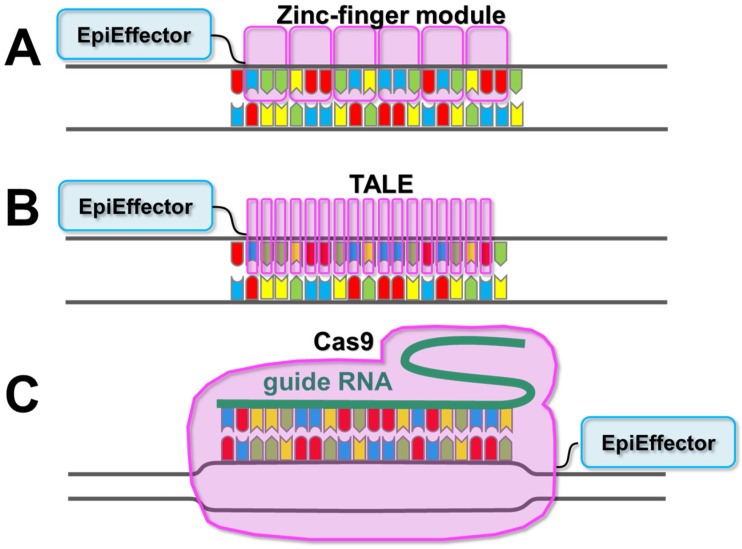
Schematic diagram of the DNA recognition domains available for epigenomic-modifying enzymes. (**A**) In zinc-finger arrays, each zinc-finger module recognizes three nucleotides. (**B**) In transcription activator-like effectors (TALEs), each repeat recognizes one nucleotide. (**C**) In clustered regulatory interspaced short palindromic repeats (CRISPR)/CRISPR-associated protein 9 (Cas9) (CRISPR/Cas9), one strand of the target site is recognized through Watson–Crick base pairing by a bound guide RNA. The attached effector domain (EpiEffector) is indicated by a blue shape. For details on EpiEffectors, please refer to Table 1 and earlier reviews [37,41,43,44,45].

**Figure 3 ijms-24-04778-f003:**
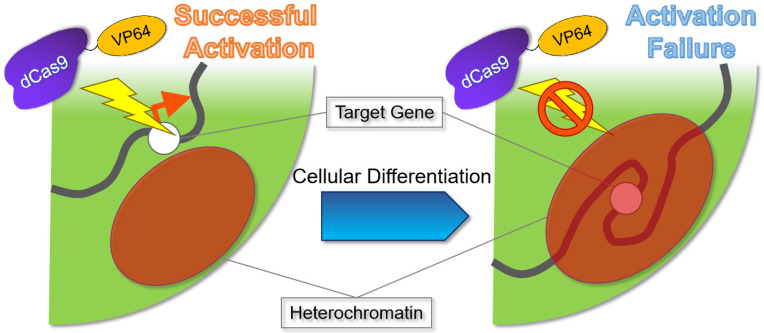
Schematic illustration of the importance of the chromatin structure in epigenome editing. During cell differentiation, the cell nucleus (light green) forms a heterochromatin (red) [137,138,141]. If the target gene is located within the heterochromatic region, it is inaccessible for expression (right panel). Dead Cas9 (dCas9)-four tandem repeats of the transcriptional activator VP16 (VP64) (dCas9-VP64) is shown as an example of an epigenomic-modifying enzyme.

**Figure 4 ijms-24-04778-f004:**
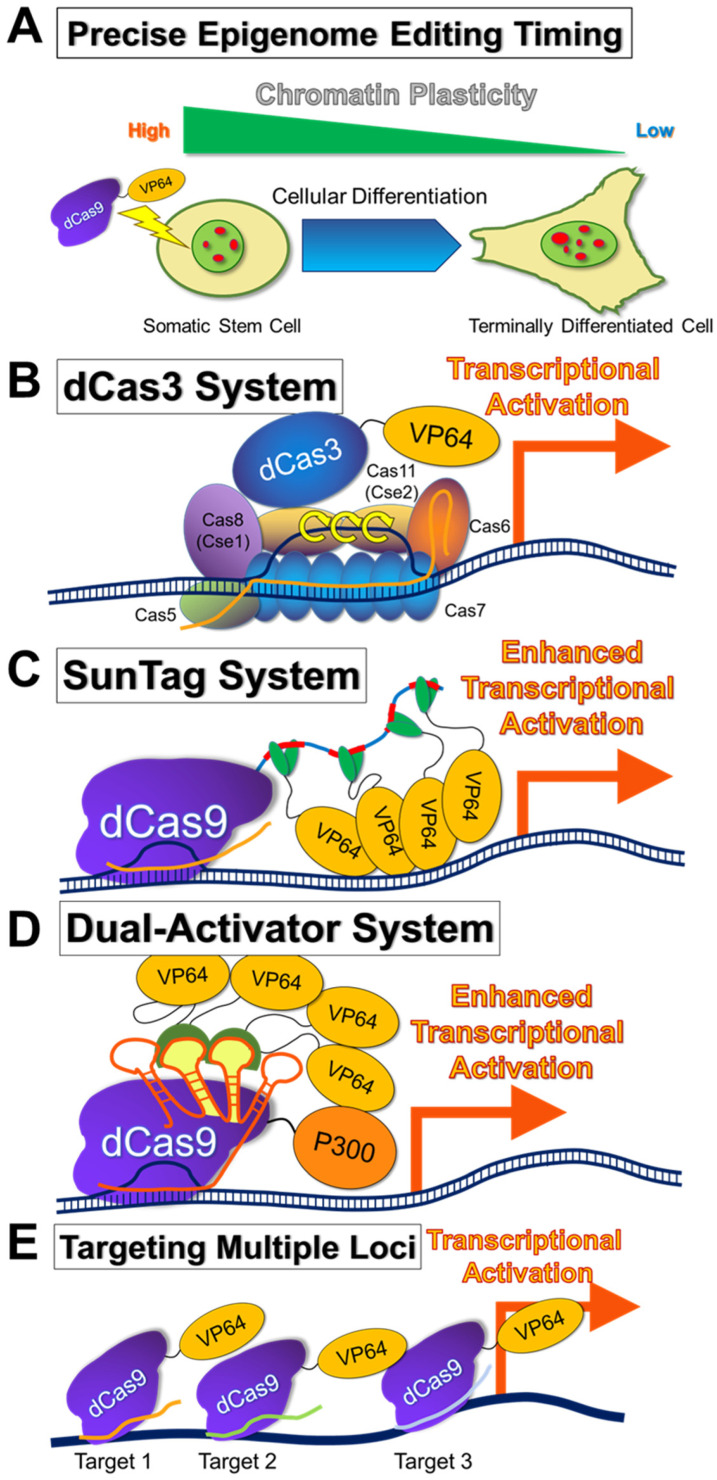
Strategies to improve therapeutic epigenome editing (**A**) Upon differentiation, the cell nuclei (light green) lose chromatin plasticity through forming heterochromatin structures (red dots) [137,141]. Chromatin plasticity must be considered for more effective and efficient epigenome editing. (**B**) Type I-E CRISPR effector is composed of CRISPR RNA (crRNA), Cas3 (possesses helicase and nuclease activity), and a large Cascade complex, which contains Cas5, Cas6, multiple Cas7, Cas8 (Cse1), recognizing the protospacer adjacent motif (PAM), and two Cas11 (Cse2) [131,133]. Although the dead Cas3 (dCas3) complex has not been used for epigenome editing, it would be important to determine its effect on epigenome editing. VP64 is shown as an example of the EpiEffector. (**C**) A scheme for CRISPR–dCas9 and a peptide repeat-based amplification of transcriptional activity using VP64 as an example of a SUperNova tag (SunTag) system. dCas9 fused with a peptide repeat can recruit multiple copies of single-chain fragment variable (scFv)-fused VP64 antibody [115]. Thus, multiple copies of VP64 can activate the target gene more efficiently. (**D**) A scheme for amplification of transcriptional activity using the dual-activator enCRISPRa system. P300 and VP64 are shown as examples of transcriptional activation EpiEffectors. VP64 is fused with MS2 coat protein (MCP), and MCP-VP64 fusion protein binds to MS2 hairpins within the single guide RNA (sgRNA) [54,115]. P300 and VP64 act synergistically to activate the target gene. CRISPRon and CRISPRoff (not shown in the figure; single artificial genes containing multiple EpiEffectors together with dCas9) are similar systems in which multiple EpiEffectors are simultaneously expressed to activate or repress a target gene [70]. (**E**) Targeting multiple loci within the target gene using epigenomic-modifying enzymes is a simple approach to effectively modulate the epigenome [98,99]. Three dCas9-VP64 targeting three different loci are shown as examples.

**Table 1 ijms-24-04778-t001:** Summary of EpiEffectors and their induced effects.

Application	EpiEffector	Induced Epigenetic Changes
H3K27 acetylation (Gene activation)	P300 [46,47,48,49,50,51,52,53,54,55] cAMP-response element binding protein (CREB)-binding protein (CBP) [56] P300 and/or CBP [57] VP64 + P300 [54] MS2-P65-HSF1 (MPH) [58]	Increase in H3K27 acetylation, H3 acetylation Enhanced expression of target genes Enhanced expression of target genes Increase in H3K4 trimethylation and H3K27 acetylation
H3K27 deacetylation	Histone deacetylase 3 (HDAC3) [59]	Decrease in H3K27 acetylation
H3K4 methylation (Gene activation)	SET and MYND domain-containing protein 3 (SMYD3) [60] PR domain zinc finger protein 9 (PRDM9) [61] Disruptor of telomeric silencing 1-like (DOT1L) [61] Ubiquitin-conjugating enzyme E2 A (UBE2A) [61] BRG1/BRM associated factor (BAF) (SS18 subunit) [62]	Increase in H3K4 methylation Increase in H3K4 trimethylation Increase in H3K79 trimethylation Loss of H3K27 trimethylation and increase in H3K4 trimethylation
H3K9 and H3K27 methylation (Gene repression)	Lysine-specific demethylase 1 (LSD1) [54,63] Krüppel-associated box (KRAB) [14,15,54,55,56,63,64,65,66,67,68,69,70] JUMONJI (JMJ) [71] G9A (also known as Euchromatic histone-lysine N-methyltransferase 2 (EHMT2)) [65] Suppressor of Variegation 3–9 Homolog 1 (SUV39H1) [65] Enhancer of zeste homolog 2 (EZH2) [65,66,68,72] Friend of GATA protein 1 (FOG1) [65,68] LSD1 + KRAB [54] heterochromatin protein 1 (HP1) [62]	Decrease in H3K9 dimethylation and H3K27 acetylation Decrease in H3K27 acetylation and increase in H3K27 trimethylation Decrease in H3K4 trimethylation Increase in H3K9 trimethylation Increase in H3K9 trimethylation Increase in H3K27 trimethylation Decrease in H3K27 acetylation and increase in H3K27 trimethylation Decrease in H3K4 mono- and dimethylation Increase in H3K9 trimethylation
DNA methylation (Gene repression)	DNMT3A [65,66,73,74,75,76,77,78,79,80,81,82,83] DNMT3A + DNMT3L [70,84,85,86] KRAB/EZH2/FOG1 + DNMT3A [65] KRAB + DNMT3A (+ DNMT3L) [47,70] M. SssI MQ1 [87,88,89,90] DNMT1 [78] DNMT3B [78]	Increase in DNA methylation Increase in DNA methylation
DNA demethylation (Gene activation)	TET1 [47,70,73,83,91,92,93,94] TET3 [95] CRISPR activation (CRISPRa) + TET1 [96]	Decrease in DNA methylation Increase in 5-hydroxymethylcytosine Decrease in DNA methylation

**Table 2 ijms-24-04778-t002:** Epigenome-editing studies in vivo.

	Platform	EpiEffectors	Species	Target Diseases	Effects	Carrier, Gene Delivery Methods	Reference
1	Zinc-finger protein	Kox-1 KRAB domain	Mouse	Huntington’s disease	Repression of mutant *htt* gene	Stereotaxic injection	[14]
2	dCas9	VP64 or three copies of transcriptional repressor domain SRDX	Arabidopsis	Not applicable	Activation or repression of target genes (activation: AtPAP1, miR319; repression: AtCSTF64, miR159A, miR159B)	Transgenic plant	[98]
3	dCas9-SunTag	scFV-TET1	Mouse	Not applicable	Demethylation of Gfap regulatory region.	Electroporation	[91]
4	dCas9	DNMT3A or TET1	Mouse	Not applicable	Demethylation of BDNF promoter or de novo methylation of CTCF motifs	Stereotaxic injection of lentivirus	[73]
5	dCas9	An engineered prokaryotic DNA methyltransferase MQ1	Mouse	Not applicable	DNA methylation of H19 locus	Microinjection of gene expressing plasmid	[89]
6	TALE	A bacterial CpG methyltransferase MQ1 (SssI)	Mouse	Not applicable	Methylation of major satellite DNA	Microinjection of mRNA into the embryo	[87]
7	dCas9 + dead sgRNA (dgRNA)	MS2-P65-HSF1 (MPH)	Mouse	Duchenne muscular dystrophy, acute kidney injury, diabetes	Activation of Klotho, Utrophin, Fst, and Pdx1	Tail vein injection of AAV9	[58]
8	Staphylococcus aureus dCas9	KRAB	Mouse	To lower low-density lipoprotein cholesterol levels	Repression of Pcsk9 expression	AAV, dual-vector AAV8 system	[67]
9	high-fidelity dCas9	TET3 catalytic domain	Mouse	Fibrosis	Activation of Rasal1 and Klotho expression	Renal artery/vein injection of lentivirus	[95]
10	CRISPR-Act2.0 and mTALE-Act	VP64	Arabidopsis	Not applicable	Activation of multiple (CSTF64, GL1, and RBP-DR1) genes	Transgenic plant	[99]
11	dCas9	TET1	Mouse	Fragile X syndrome	Activation of FMR1 expression	Epigenome-edited neural precursor cells were injected into the brain	[92]
12	dCas9	VP64	Mouse	Obesity	Activation of Mc4r expression	Stereotaxic injection of AAV-DJ	[100]
13	dCas9	VP64	Mouse	Muscular dystrophy	Activation of Lama1 expression	Tail vein injection of AAV9	[101]
14	dCas9	DNMT3A or TET1	Mouse	Not applicable	Repression or activation of A^vy^ locus	Microinjection	[83]
15	dCas9	Oryzias latipes EZH2	Medaka	Not applicable	H3K27 methylation of Arhgap35, Nanos3, Pfkfb4a, Dcx, Tbx16, and Slc41a2a	Injection of mRNA	[72]
16	dCas9-SunTag	scFv-C11orf46	Mouse	Hypoplasia of the corpus callosum	Normalization of Sema6a expression	In utero electroporation	[102]
17	Zinc-finger protein	KRAB	Mouse	Huntington’s disease	Repression of mutant *htt*	Stereotaxic injection of AAV2/6 or AAV2/9	[15]
18	dCas9	VP64	Mouse	Dravet syndrome	Activation of Scn1a expression	Intracerebroventricular injection of AAV9	[103]
19	dCas9	VPR	Mouse	Blindness	Activation of Opn1mw expression	Dual adeno-associated viral vectors	[104]
20	dCas9-SunTag	TET1 catalytic domain	Mouse	Generation of Silver–Russell syndrome disease model	Demethylation of H19-DMR and repression of Igf2	Microinjection of mRNA, transgenic mice	[93]
21	dCas9-SunTag	scFv-TET1 catalytic domain	Mouse	Not applicable	Activation of Fgf21 expression	Hydrodynamic tail vein injection	[94]
22	enCRISPRi	LSD1 and KRAB	Mouse	Not applicable	Perturbation of enhancers during hematopoiesis	Tetracycline-inducible knock-in mice	[54]
23	Staphylococcus aureus dCas9	KRAB	Mouse	To lower low-density lipoprotein cholesterol levels	Repression of Pcsk9 expression	Tail vein injection of AAV8	[69]
24	dCas9	A bacterial CG-specific DNA methyltransferase MQ1 Q147L	Arabidopsis	Not applicable	Repression of FWA expression	Transgenic plant	[90]
25	dCas9	P300 or KRAB	Rat	Alcohol use disorder	Activation or repression of Arc expression	Stereotaxic injection of lentivirus	[55]
26	dCas9	VP64, JMJ	Arabidopsis	Not applicable	Repression of APX2 expression	Transgenic plant	[71]

## Data Availability

No new data were created or analyzed in this study. Data sharing is not applicable to this article.

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
