# Peer review of "Toward the Development of Epigenome Editing-Based Therapeutics: Potentials and Challenges"

_ijms, 2023, doi:10.3390/ijms24054778_

Round 1

Reviewer 1 Report

This review manuscript “Toward the development of epigenome editing-based therapeutics: potentials and challenges” is an interesting topic. The manuscript highlights the emerging epigenome editing therapeutics in vivo. The manuscript is well written. The scientific content of the manuscript merits for publication. However, the following concerns need to be addressed:

1.     The authors may include Immunogenicity in challenges in epigenome editing technologies.

2.     The authors may consider including the ethical concerns in the use of these technologies. 

Author Response

This review manuscript “Toward the development of epigenome editing-based therapeutics: potentials and challenges” is an interesting topic. The manuscript highlights the emerging epigenome editing therapeutics in vivo. The manuscript is well written. The scientific content of the manuscript merits for publication. However, the following concerns need to be addressed:

1. The authors may include Immunogenicity in challenges in epigenome editing technologies.

Thank you for your important and valuable suggestions to improve the manuscript. We agree that epigenome editing technologies, both CRISPR-Cas and TALE systems, are potentially immunogenic in that they contain non-human material. In this regard, we have added the following sentences in the text (line 224-229).

More so, epigenome editing technologies using CRISPR-Cas and TALE are potentially immunogenic in that they contain non-human materials (Crudele et al., Nature Communications, 2018). In contrast, zinc-finger-based technologies have the advantage of being less immunogenic than TALE and CRISPR because they are composed of polypeptides encoded in the human genome. In any case, one of the challenges for the future will be how to reduce the immunogenicity of the components of epigenome editing technologies.

2. The authors may consider including the ethical concerns in the use of these technologies.

Thank you for your important and valuable suggestions to improve the manuscript. Accordingly, we have added the following texts in the section “6. Future perspectives” (line 478-483).

Finally, because epigenome editing does not involve modification of the genome itself, it is currently considered to have a lower impact on germ cells than genome editing. Thus, epigenome editing has the potential to overcome important scientific and ethical issues of concern with genome editing. However, because of the uncertainties associated with new medical technologies, deliberation is essential on how to clear social and ethical issues and develop safe and appropriate strategies and policies (Zeps et al., Stem Cell Reports, 2021).

Reviewer 2 Report

Ueda et.al. reviewed the application of epigenome editing for the potential treatment of various diseases. The work is very interesting and the figures illustrate the process very well and help readers understand the mechanisms. I recommend proofreading the manuscript (for example, line 75, K3K27). I recommend the review be published.

Author Response

Ueda et.al. reviewed the application of epigenome editing for the potential treatment of various diseases. The work is very interesting, and the figures illustrate the process very well and help readers understand the mechanisms. I recommend proofreading the manuscript (for example, line 75, K3K27). I recommend the review be published.

Thank you very much for your excellent effort and suggestion concerning line 75, K3K27. We revised the sentence (line 75, K3K27) as follows (line 74-75).

In contrast, methylation of the 9th or 27th lysine residue of the same H3 (H3K9 or H3K27) is involved in transcriptional repression and heterochromatin structure formation [27-29].

Reviewer 3 Report

The manuscript is coherently written, with a thorough introduction which covers a wide range of topics for the development of epigenome editing-based therapeutics. However, the following suggestions might improve the manuscript's quality (Manuscript ID: IJMS-2183457).

The following suggestions are:

Author should have mentioned in “Challenges in epigenome-editing technologies section” that limitations of targeting EpiEffector for alternative splicing of a gene. When EpiEffector targets any gene by 20 nucleotide gRNA, which will affect locally eithor one of the exon and gene can still can be transcribe via alternative splicing mechanism, which might be one of the limitations using EpiEffector technology.

Similarly, Author, should have emphasis in “Challenges in epigenome-editing technologies section” how long term effect of EpiEffector technology might help. As we know that epigenetic is reversible, what if a gene needs to be suppressed or activated for longer time, how EpiEffector approach is beneficial.

Line # 129, the word “incapable” is not making any sense, author need to correct as any appropriate word like non-specific or other.

Also, Author mentioned about chromatin plasticity and had citied some of research articles, however, author have not citied this article https://doi.org/10.1016/B978-0-12-817819-5.00008-5” and it in Genome Plasticity in Health and Disease book chapter 8, which states how epigenetics and factors involved in cellular chromatin plasticity and are important players in human health and disease.

Author Response

The manuscript is coherently written, with a thorough introduction which covers a wide range of topics for the development of epigenome editing-based therapeutics. However, the following suggestions might improve the manuscript's quality (Manuscript ID: IJMS-2183457).

The following suggestions are:

Author should have mentioned in “Challenges in epigenome-editing technologies section” that limitations of targeting EpiEffector for alternative splicing of a gene. When EpiEffector targets any gene by 20 nucleotide gRNA, which will affect locally either one of the exon and gene can still be transcribe via alternative splicing mechanism, which might be one of the limitations using EpiEffector technology.

Thank you for your excellent and valuable suggestion. Accordingly, we added the texts as follows (line 280-284).

However, when EpiEffector targets a transcriptional regulatory region of about 20 nucleotides, it should be noted that the effect of epigenome editing may become local, and the gene may be transcribed normally through alternative splicing and alternative promoter mechanisms (Kornblihtt et al., Nat Rev Mol Cell Biol., 2013). Accordingly, the possibility of such splicing should be considered when applying this technology.

Similarly, Author, should have emphasis in “Challenges in epigenome-editing technologies section” how long-term effect of EpiEffector technology might help. As we know that epigenetic is reversible, what if a gene needs to be suppressed or activated for longer time, how EpiEffector approach is beneficial.

Thank you for pointing this out. As described in line 322-323, the advantage of epigenome editing is that repeated dosing is possible. Your point is important, so we have added the following texts (line 451-454).

In addition, since epigenetics are reversible, repeated administration of epigenome-editing enzymes may be necessary if the target gene needs to be repressed or activated for an extended period. Accordingly, a dosage form that allows repeated administration is also desirable and should be taken into consideration.

Line # 129, the word “incapable” is not making any sense, author need to correct as any appropriate word like non-specific or other.

Thank you for pointing this out. Accordingly, we have revised the text as follows (line 127-129).

EpiEffectors are enzyme domains of a group of enzymes involved in the epigenetic modification of DNA and histone proteins that do not themselves bind to specific DNA sequences.

Also, Author mentioned about chromatin plasticity and had citied some of research articles, however, author have not citied this article https://doi.org/10.1016/B978-0-12-817819-5.00008-5” and it in Genome Plasticity in Health and Disease book chapter 8, which states how epigenetics and factors involved in cellular chromatin plasticity and are important players in human health and disease.

Thank you for your suggestion. We have added the following to the references you provided (line 301-303).

The plasticity of the nuclear structure (genomic organization) of a cell is lost as cell differentiation occurs, and certain diseases are known to have abnormal nuclear structures [136-139].
